The avian fossil record in Insular Southeast Asia and its implications for avian biogeography and palaeoecology

Meijer Hanneke J.M. Hanneke.Meijer@icp.cat
Institut Català de Paleontologia Miquel Crusafont , Cerdanyola del Vallès , Spain
Human Origins Program, National Museum of Natural History, Smithsonian Institution , Washington, DC , USA
Naturalis Biodiversity Center , Leiden , The Netherlands
Hutchinson John
Electronic publication date: 2014 Mar 11
Publication date: 2014
Volume: 2
Electronic Location ID: e295
Received 2014 Jan 21; Accepted 2014 Feb 7
Copyright: © 2014 Meijer
Copyright year: 2014
Copyright holder: Meijer
License: This is an open access article distributed under the terms of the Creative Commons Attribution License, which permits unrestricted use, distribution, reproduction and adaptation in any medium and for any purpose provided that it is properly attributed. For attribution, the original author(s), title, publication source (PeerJ) and either DOI or URL of the article must be cited.
License URL: https://creativecommons.org/licenses/by/4.0/

Keywords: Insular Southeast Asia, Dispersal, Wallacea, Extinction, Paleontology, Fossil birds, Avian biogeography

Funding: Spanish Ministerio de Economía y Competitividad CGL2011-28681 Generalitat de Catalunya BP-B-00174 This work has been supported by the Spanish Ministerio de Economía y Competitividad (CGL2011-28681), and the Generalitat de Catalunya (BP-B-00174 to HJMM). The funders had no role in study design, data collection and analysis, decision to publish, or preparation of the manuscript.

==============================
Excavations and studies of existing collections during the last decades have significantly increased the abundance as well as the diversity of the avian fossil record for Insular Southeast Asia. The avian fossil record covers the Eocene through the Holocene, with the majority of bird fossils Pleistocene in age. Fossil bird skeletal remains represent at least 63 species in 54 genera and 27 families, and two ichnospecies are represented by fossil footprints. Birds of prey, owls and swiftlets are common elements. Extinctions seem to have been few, suggesting continuity of avian lineages since at least the Late Pleistocene, although some shifts in species ranges have occurred in response to climatic change. Similarities between the Late Pleistocene avifaunas of Flores and Java suggest a dispersal route across southern Sundaland. Late Pleistocene assemblages of Niah Cave (Borneo) and Liang Bua (Flores) support the rainforest refugium hypothesis in Southeast Asia as they indicate the persistence of forest cover, at least locally, throughout the Late Pleistocene and Holocene.

Introduction

Ever since the seminal observations of Wallace (1863) and Wallace (1869), a plethora of studies has highlighted the avian species richness, high levels of endemism, and unique biogeographical patterns, as well as the conservation threats across Insular Southeast Asia (i.e., Huxley, 1868; Mayr, 1944; Butchart et al., 1996; Brooks, Pimm & Collar, 1997; Brooks et al., 1999; Stattersfield et al., 1998; Myers et al., 2000; Trainor & Lesmana, 2000; Birdlife International, 2001; Jones et al., 2001; Jones, Marsden & Linsley, 2003; Ding et al., 2006; Jønsson et al., 2008; Jønsson et al., 2010; Holt et al., 2013). However, very little is known regarding the evolutionary history of birds in this region. Evidence for extinction events and faunal turnovers in Insular Southeast Asia comes mainly from the mammalian fossil record, particularly proboscideans and primates (i.e., van den Bergh, 1999; Meijaard, 2003; de Vos, van den Hoek Ostende & van den Bergh, 2007; Van der Geer et al., 2010; Louys & Meijaard, 2010). How past events have shaped current patterns of avian distributions remains unexplored. Here, I provide a synthesis of the current fossil record of birds in Insular Southeast Asia. I describe sites with fossil avian remains, interpret the fossil evidence in terms of avian biogeography and dispersal, and discuss the significance of avian assemblages for Southeast Asian palaeoecology.

Material and Methods

Data on fossil occurrences of avian taxa in Insular Southeast Asia were acquired through the database www.Fossilworks.org (search terms ‘Aves’ and ‘Asia’), from published accounts, and from my own observations. Insular Southeast Asia is defined as the islands in the archipelagos of Indonesia, the Philippines, Malaysia, Singapore, Brunei, East Timor, and Christmas Island.

Localities with avian remains in Insular Southeast Asia

Although Australia and Asia have both yielded extraordinary fossil bird remains (for an overview, see Murray & Vickers-Rich (2004) and Chiappe (2007)), some of them dating back well into the Cretaceous and including some of the world’s earliest birds (Vickers-Rich, 1991; Chiappe, 2007; Martin et al., 2014), fossil bird remains from the island archipelagos in between these continental landmasses are rare. There are no Mesozoic bird remains, which can be attributed to the fact that the majority of islands didn’t constitute dry land until the Cenozoic (Hall, 2002).

Sumatra

The oldest bird remains from Insular Southeast Asia are represented by an almost complete skeleton of the pelecaniform Protoplotus beauforti (Lambrecht, 1931) from Sibang in the Sangkarewang Formation, West Sumatra (Fig. 1). The age of the sediments has been disputed, but they are thought to be at least of Eocene age (Van Tets, Rich & Marino-Hadiwardoyo, 1989). Avian footprints have been described from the Oligocene Sawahlunto Formation near Kandi in West Sumatra. These footprints are attributed to two distinct ichnospecies of Aquatilavipes, A. wallacei (Zonneveld et al., 2011) and Aquatilavipes sp. A (Zaim et al., 2011; Zonneveld et al., 2011; Zonneveld et al., 2012), and are most similar to tracks produced by small shorebirds such as sandpipers, rails and plovers (Zonneveld et al., 2011).

Figure 1 Map of Insular Southeast Asia indicating the location of localities with fossil bird remains.

Java

Weesie (1982) described avian remains collected from the Middle Pleistocene site of Trinil HaubtKnochen (HK), the type locality for Homo erectus (Dubois, 1893), and from Sumber Kepuh, a locality that is considered to be contemporaneous with H. erectus (Storm, 2012). Weesie (1982) ascribes the fossils from Trinil HK to the greater adjutant Leptoptilos cf. dubius (Gmelin, 1789), black-necked stork Ephippiorhynchus cf. asiaticus (Latham, 1790), the red-breasted goose Branta cf. ruficollis (Pallas, 1769), and the Australian shelduck Tadorna tadornoides (Jardine & Selby, 1828) and the single fossil from Sumber Kepuh to the green peafowl Pavo m. muticus (Linnaeus, 1766) (Table S1).

Wetmore (1940) described bird bones from bone-bearing terraces near Watualang, situated near the Solo River in central Java (Fig. 1). These terraces are considered to be Late Pleistocene of age and identical with the Ngandong terraces. The bones represented two fossil bird species: a crane, Grus grus (Linnaeus, 1758) and a new species of giant stork, Leptoptilos titan (Wetmore, 1940). Wetmore (1940) also tentatively mentions remains of an extinct vulture of the subfamily Aegypinae, but the whereabouts of these fossils is unknown. Bird remains have also been reported from Wajak Cave, East Java (van den Brink, 1982). The bones are ascribed to an owl, Strigiformes indet., and a songbird, Passeriformes indet. The ages suggested for Wajak have ranged from Pleistocene to Holocene, but Storm et al. (2012) found that at least part of the assemblage is Late Pleistocene in age.

Philippines

Reis & Garong (2001) reported on an Early Holocene terrestrial vertebrate assemblage excavated in four caves in Quezon Municipality on Palawan. Thirty-five bones were identified as avian, and could be assigned to twelve avian taxa (Table S1); a species of scops owl, Otus sp., three species of swiftlets in the genus Collocalia (Gray, 1840) (note that C. salangana (Streubel, 1848) is here treated as Aerodramus salanganus following Chantler, Wells & Schuchmann, 1999), and eight species of passerine birds.

Borneo

Stimpson (2009), Stimpson (2010) and Stimpson (2013) described fossil bird assemblages associated with human settlements from the Late Pleistocene and Holocene sediments of Great Cave of Niah in Sarawak, North-western Borneo. He identified 28 taxa (Table S1) in 8 families, including raptors (Accipitridae) and owls (Strigidae), hornbills (Bucerotidae), and swiftlets (Apodidae).

Flores

Late Pleistocene and Holocene bird remains from Liang Bua, a limestone cave on the western part of the island, were described by Meijer & Due (2010) and Meijer et al. (2013). The Liang Bua fossil bird assemblage is diverse and contains at least 26 non-passerine taxa in 13 families (Table S1), including a giant marabou stork Leptoptilos robustus (Meijer & Due, 2010), a vulture Trigonoceps sp., owls, pigeons (Columbidae), parrots (Psittacidae) and swiftlets (Apodidae).

Other sites

Bird remains have been documented for Madai Cave on Sabah (Harrison, 1998) and Early/Middle Pleistocene sites in the Soa Basin of Central Flores (Brumm et al., 2006) but these remains have not been identified yet.

The Fossil Bird Record in Insular Southeast Asia

To date, sixty-three species of fossil birds have been identified in Insular Southeast Asia, representing at least 54 genera and 27 families (Table S1). Two taxa are represented by fossil footprints only. While the avian fossil record covers the Eocene through to the Holocene, the majority of fossil avians come from Quaternary sediments. Raptors, owls, and swiftlets are present and abundant in all cave sites excavated, and remain ubiquitous elements in the modern Southeast Asian avifauna today. The current fossil record for birds forms a significant extension of the last overview of Southeast Asian fossil birds (Rich et al., 1986), which only listed a record for Java. The recent expansion reflects both renewed interest in existing collections (Borneo) as well as new excavation efforts on Sumatra, Flores and Palawan.

It is important to note the differences in taphonomy between some of the sites. Open-air sites in fluvial deposits, such as the localities of Watualang and Trinil, are less likely to preserve microfauna, whereas the cave sites on Borneo, Palawan and Flores are more favourable for the preservation of smaller bones. Bone accumulations in caves may result from fluvial transport, pitfalls, burrow deaths, predation and hibernation/aestivation (Andrews, 1990). For Niah Caves, deposition of swiftlets likely resulted from natural death, as the caves have sustained populations of swiftlets for at least 48,000 years (Stimpson, 2013). At Liang Bua (Flores), signs of digestions on the bone surfaces suggest that birds of prey, most likely a species of barn owl (Tyto sp.), are responsible for the accumulation of the majority of non-predatory birds (Meijer et al., 2013). Although many bird bones have been found in association with hominid (Java, Flores) or modern human (Borneo and Palawan) remains, no cut marks or other signs indicative of human hunting have been reported for any bird bones, in contrast to the remains of the pygmy elephant Stegodon florensis insularis (van den Bergh et al., 2008; van den Bergh et al., 2009).

Discussion

Avian biogeography

Avian dispersal patterns in Insular Southeast Asia are complex (Jønsson et al., 2008; Jønsson et al., 2010; Carstensen & Olesen, 2009; Michaux, 2010), as the geological history of the region is complicated (Hall, 2002) and the avifauna contains elements of both Indo-Malayan and Australasian origin (Mayr, 1944; Michaux, 2010). Especially for the Philippines (minus Palawan) and Wallacea, the use of land bridges to explain inter-island faunal similarities is limited, as these archipelagos were never connected to mainland Asia or Australia. The deep sea-straits between islands formed a dispersal barrier for terrestrial animals, and although birds are generally less hindered by such barriers, the extent to which sea straits affected avian dispersal and diversification in Insular Southeast Asia is poorly understood (Hosner, Nyári & Moyle, 2013). Molecular studies suggest complex patterns of mainland-island interchange and diversification for passerines (Cumings Outlaw & Voelker, 2008; Jønsson et al., 2010; Jønsson et al., 2011), but little work has been done for non-passerine birds (Birks & Edwards, 2002). Fossil occurrences can provide calibration points for dispersal models based on molecular data, and can reveal past dispersal routes and events that may go unnoticed when only extant taxa are considered.

With the exception of the extinct giant storks from Flores and Java (Table S1), species recorded in fossil assemblages are extant species, suggesting continuity of avian lineages across Southeast Asia since at least the Late Pleistocene. A contraction in species range does seem to have occurred for a number of species. The greater adjutant and the red-breasted goose from the Middle Pleistocene of Java are currently restricted to more northern regions. The Australian shelduck and the black-necked stork are nowadays occasional visitors, and Weesie (1982) considers their presence in Middle Pleistocene Java as indicative of cooler climatic conditions. Whereas the Middle Pleistocene mammal fauna from Java went extinct at the transition to the Late Pleistocene (de Vos, van den Hoek Ostende & van den Bergh, 2007; Van der Geer et al., 2010), birds might have responded by a range shift rather than extinction. On Palawan, the mountain white-eye (Zosterops montanus Bonaparte, 1850) now occurs at higher elevations and seems to have undergone a recent range contraction (Reis & Garong, 2001). The presence of vultures in Late Pleistocene Insular Southeast Asia is significant, as they are conspicuously absent from the modern avifauna (Thiollay, 1998). Remains of the white-headed vulture Trigonoceps sp. were recovered from Late Pleistocene sediments at Liang Bua (Flores) (Meijer et al., 2013). Wetmore (1940) reported on two bones from Late Pleistocene sediments at Watualang (Java) “being from a species about the size of Pseudogyps bengalensis. These are typical in form of birds of this group and apparently represent an extinct species” (Wetmore, 1940, p. 450). Unfortunately, the whereabouts of these bones remains unclear and their vulturine nature cannot be confirmed at present. The presence of vultures on Flores, and possibly on Java, indicates that the ranges of extant genera of vultures, even those with current distributions limited to Africa, may have been much larger in the past, a conclusion also supported by the presence of two African vulture genera in the middle Pleistocene of China (Zhang et al., 2012). According to Thiollay (1998), the lack of vultures in the modern avifauna results from the lack of mammalian carcasses on islands. Alternatively, species impoverishment, and with that the lack of certain species, along the continent-island gradient can results from nestedness, in which only the abundant, generalist and forest species make up the poorer, island species subset. Avifaunas within Southeast Asia indeed show a high level of nestedness (Carstensen & Olesen, 2009), but with evidence for fossil vultures in Southeast Asia, nestedness does not explain the modern absence of vultures. The disappearance of Trigonoceps sp. from the Liang Bua sequence at the end of the Late Pleistocene seems to be tied to the disappearance of Homo floresiensis (Brown et al., 2004) and Stegodon florensis insularis (Meijer et al., 2013), which left the island devoid of large mammals. Also, both the extant as well as the fossil Javan fauna contains a number of large mammals, including large bovids and deer (and until recently, proboscideans and rhinos as well), which could have provided potential food resources for vultures. The current absence of vultures in insular Southeast Asia seems to be an effect of Late Pleistocene extinction events rather than nestedness.

Avian Dispersal Routes

For the Lesser Sunda Islands, different dispersal scenarios have been proposed. Stresemann (1939) suggested that certain terrestrial birds colonized the Sunda Islands from the north (Sulawesi), but others have argued for a route via Southeastern Sundaland based on the similarities between the extant avifaunas of Flores, Java and Bali (Mayr, 1944; Mees, 2006). Such a route agrees with the presence of giant storks in the Late Pleistocene of both Java and Flores, as their large size might represent an adaptation to a more terrestrial life style (though it should be noted here that L. robustus from Flores displays wing morphology similar to modern L. dubius, Meijer & Due, 2010). Both the Javan and Flores giant stork may represent late offshoots of the lineage that also contains L. falconeri (Milne-Edwards, 1867–1871) from the Siwalik Hills in India (Louchart et al., 2005). Living Ciconidae display significant sexual dimorphism (Louchart et al., 2005), but Weesie (1982) does not consider the possibility that the stork remains from the Middle Pleistocene of Java (which he attributed to the modern black-necked stork Ephippiorhynchus cf. asiaticus and the greater adjutant L. cf. dubius) might, in fact, be smaller (female) specimens of the giant stork L. titan.

Although most of the Sunda Shelf was dry land at some point during the Pleistocene (Hall, 2002), there is no unambiguous geological evidence that Palawan was once connected to Sundaland (see discussion in Reis & Garong, 2001). Fossil and modern faunal data suggest that Palawan might have been a stepping stone between Borneo and the Philippines. In that light, the fossil presence of the mossy-nest swiftlet Aerodramus cf. salanganus in the very early Holocene is interesting, since it is currently absent on Palawan, but present in the Greater Sundas as well as in the Philippines. Its presence in the early Holocene could indicate that Palawan served as a stepping-stone for the dispersal of A. salanganus from the Greater Sundas into the Philippines.

Palaeoecological Implications

The climatic and ecological conditions of Pleistocene Sundaland have been subject of much debate. Heaney (1991) proposed the presence of a continuous savannah corridor during glacial maxima, which stretched from the Malaysian Peninsula through central Sundaland all the way to the Lesser Sunda Islands. This savannah corridor presumably facilitated the dispersal of hominins and other terrestrial megafauna across Sundaland, and acted as a dispersal barrier for forest-dependent taxa between Sumatra and Borneo. Although geomorphological, biogeographical, and palynological evidence, as well as vegetation modelling indicate that drier conditions prevailed over Insular Southeast Asia during the Last Glacial Period (for a review, see Bird, Taylor & Hunt, 2005), there is only very limited support for a continuous savannah corridor at the scale proposed by Heaney (1991). The savannah corridor hypothesis is based on the climatic conditions that occurred during glacial maxima, a situation that existed only for short periods of time during the Pleistocene (Voris, 2000). Furthermore, Slik et al. (2011) argue that the biogeographic difference between Sumatra and the Malay Peninsula, and Borneo can also be explained by exposed sandy sea-bed soils that acted as a dispersal barrier. Instead of savannah vegetation, swamps and heath forest dominated central Sundaland.

Birds are closely associated with vegetation, and their abundance and diversity in fossil assemblages adds valuable palaeoecological insights. The habitat spectrum inferred from Liang Bua’s avifauna (Meijer et al., 2013) shows that, during the Late Pleistocene, forest habitats, as well as wetland and open grassland habitats dominated around Liang Bua, and provided ample resources for hominins. Studies of the fossil bird assemblage from Niah Cave on Borneo (Stimpson, 2010; Stimpson, 2013) showed that swiftlets, obligate insectivores, had been persistently present from well into the Late Pleistocene until today (Stimpson, 2013). Their continuous presence indicates that habitats within the feeding range of these birds supported a sufficient base of aerial arthropods to support swiftlet populations. Despite ample evidence for drier conditions based on geomorphological, palynological, biogeographical data and climate modelling (see references in i.e., Bird, Taylor & Hunt, 2005; Westaway et al., 2009), and the concomitant contraction of forests, Late Pleistocene fossil bird assemblages from both Flores and Borneo suggest that forests persisted at least locally. These findings are consistent with other lines of evidence (Cannon, Morley & Bush, 2009; Wurster et al., 2010; Stimpson, 2012) that suggest that forest habitats were maintained across Insular Southeast Asia throughout the Late Pleistocene and Holocene.

Conclusion

The avian fossil record in Insular Southeast Asian has significantly increased in abundance and diversity over the last decades, and now consists of at least 63 species, representing 54 genera in 27 families, and 2 ichnospecies. Fossil bird remains span the last 50 million years, from the Eocene to the Holocene, but most remains have been found in Quaternary sediments. Swiftlets, birds of prey and owls are present and abundant in all cave sites excavated, and are ubiquitous elements in the modern Southeast Asian avifauna.

Extinctions seem to have been few, as the majority of the species in the fossil record are extant taxa. A number of species have undergone range shifts and are currently restricted to cooler regions. Similarities between the Late Pleistocene avifaunas of Flores and Java suggest a dispersal route across southern Sundaland. The mossy-nest swiftlet A. salanganus might have dispersed into the Philippines via Palawan. The Late Pleistocene record of forest birds, particularly swiftlets, suggest that the structural diversity of forest habitats was maintained, at least locally, throughout the Late Pleistocene and into the present.

Supplemental Information

Table S1 Fossil bird species from Insular Southeast Asia

† indicates an extinct species; (†) indicates a possibly extinct species; * indicates a species that is no longer present in Insular Southeast Asia.

Click here for additional data file.

This paper was inspired by the 2nd Southeast Asian Gateway Evolution Meeting, held March 11–15 2013, in Berlin, Germany, and benefitted from discussions with T Sutikna, RA Due, WE Saptomo, HF James, MW Tocheri, MJ Morwood, GD van den Bergh, A Brumm, M Spitzer, and C Stimpson. AAE van der Geer provided helpful feedback on an earlier version of this manuscript.

Additional Information and Declarations

Competing Interests

Author Contributions

Data Deposition

The author declares there are no competing interests.

Hanneke J.M. Meijer conceived and designed the experiments, performed the experiments, analyzed the data, contributed reagents/materials/analysis tools, wrote the paper, prepared figures and/or tables, reviewed drafts of the paper.

The following information was supplied regarding the deposition of related data:

www.fossilworks.org

13738; 68089; 74620; 77490; 106280; 142350.

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
