# Peer review of "The avian fossil record in Insular Southeast Asia and its implications for avian biogeography and palaeoecology"

_PeerJ, doi:10.7717/peerj.295_

## Round 0.1 · original submission · Minor Revisions

Congratulations on having only minor revisions to do- the reviewers really liked the paper!

Please complete these soon and I can then accept the paper- I see nothing that would require further review.

However I agree with the first reviewer that you need to clarify methods/data, sharing as much of the data involved in the study as possible: "However, there is no explicit reference in the article on methods of collection or to any databases that were consulted (e.g. Fossilworks?). This should be included in a short Materials and Methods section, following the specification of research aims (above) to show how these data were collected and to ensure reproducibility."

·

Basic reporting

The article is clearly written with a few, typos and corrections. The introduction and background are sufficient and, as this paper is a synthesis, the literature is appropriately referenced in the text and supplementary materials (but see comments under experimental design). There is scope for expansion and clarification in some instances.

General points
In general, the use of binomial and vernacular names is a little inconsistent. In some cases, one or the other is used, or both (see annotated pdf). I recommend that vernacular and binomial is used at the first mention e.g.
mossy-nest swiftlet (Aerodramus salanganus)
and any following reference to use the vernacular only (more accessible for the non-specialist)
e.g. mossy-nest swiftlet etc.
Following zoological convention, vernacular names should not be capitalised (as they are in some, but not all instances here (see annotated pdf) unless they are a noun e.g
red-breasted goose not Red-breasted Goose
Australian shelduck not Australian Shelduck
References to taxonomic authors are also generally not present in the text.
In the review of fossil sites, many bird families are referred to in the vernacular e.g. raptors, owls, swiftlets etc . In the discussion of avian biogeography that follows this, only taxonomic families names are given: Accipitridae, Strigidae, Apodidae. This may be confusing for the non-specialist reader.

Citations
There are instances of citations out of chronological order (lines 11, 17, 98), although I assume this will be dealt with in the editorial process.
Line 33: Citation: van Tet et al. 1986 in text. 1989 in references (line 353).
Line 343: correct “Palaeogreography” to “Palaeogeography”

Text
Line 57: “three species of swiftlets in the genus Collocalia”: later in the article, the mossy-nest swiftlet identified from Palawan is referred to as Aerodramus salanganus, and in the supplementary material as “Aerodramus salangana”. Perhaps re-word “three species of swiftlet: one in the genus Aerodramus and two in the genus Collocalia” and change “salangana” to “salanganus” in the supplementary table.
Line 78: delete “owls”
Line 110: “Especially the occurrence of extinct taxa may reveal successful colonization events, and thus past dispersal routes, while diminishing the (artificial) importance of other lineages”. This sentence would benefit from re-working and clarification for the non-specialist. Firstly, the mention of “successful colonization events” in the context of taxa that are now extinct may be confusing. Secondly, it is not clear which “other lineages” (i.e. extant taxa? Birds? Mammals?) are “artificially” important. Why and in what way?
Line 191: “Their continuous……indicates”. Insert “presence”
Line 197: “persisted at least locally”: Stimpson, 2012 provides a more robust case for local persistence of forest around Niah than the swiftlet article (Stimpson, 2013). In the interests of a balanced discussion, see also Hunt et al. 2012 for an alternative viewpoint on Pleistocene palaeoenvironments of Niah (references below).

Hunt, C.O., Gilbertson, D.D., Rushworth, G., 2012. A 50,000 year record of late Pleistocene tropical vegetation and human impact in lowland Borneo. Quaternary Science Reviews 27, 61-80.

Stimpson, C. M. 2012. Local-scale, proxy evidence for the presence of closed canopy forest in North-western Borneo in the late Pleistocene: bones of Strategy I bats from the archaeological record of the Great Cave of Niah, Sarawak. Palaeogeography, Palaeoclimatology, Palaeoecology 331/332, 136-149.

Supplementary material
Change “salangana” to “salanganus” (two instances)
Change “Reiss” to “Reis”

Experimental design

The article would benefit if the specific research aims and questions were “spelt out” in the introduction at line 20. These should be specified, i.e.
1) Sites and taxa are described

2) Synthesis of the implications of fossil evidence for avian biogeography and dispersal routes specifically

3) Palaeoecological implications

The literature that this article draws on is appropriately referenced in the text and supplementary materials. However, there is no explicit reference in the article on methods of collection or to any databases that were consulted (e.g. Fossilworks?). This should be included in a short Materials and Methods section, following the specification of research aims (above) to show how these data were collected and to ensure reproducibility.

Validity of the findings

All data that is described is provided in the supplementary table. Much of the article reads as speculative (though this is perhaps implicit in the title?) and is peppered with "may" or "seems" to indicate/suggest. However, I interpret this as a measured approach to drawing inferences from fossil data.

Additional comments

This review succeeds in summarising the current avian fossil record of Southeast Asia and via synthesis seeks to identify trends in avian biogeography and palaeoecology as far as the available data will permit. Discussion of these trends reads as rather speculative (arguably this is implicit in the title of the article) and the text is peppered with phrases such as "may" or "seems" to indicate/suggest - but this is interpreted as a measured approach to drawing inferences from fossil data. There are a few typos and issues with the structure and content of the article that require re-working either for clarification, to follow convention or to meet the requirements of the journal (an annotated pdf of the manuscript is attached and see comments under basic reporting, experimental design and validity of findings.)

Reviewer 2 ·

Basic reporting

No Comment

Experimental design

No comment

Validity of the findings

No comment

Additional comments

This paper gives for the first time a good overview and review of the fossil birds in Southeast Asia. The paper is well organized. Has the most recent literature. The discussion of the chapters: avian biogeography , avian dispersal and palaeoecological implications are convincing. The conclusions are in line with the discussion. To be short: a good, well organized paper and a valuable contribution to the field of science of fossil birds in Southeast Asia. Further no comments

---

## Round 0.2 · accepted · Accept

Well done! I am convinced that this now should be published.